# Physiological and Biochemical Responses of *Melilotus albus* to Saline and Alkaline Stresses

**Zhao Wang** [1,†], **Jia You** [2,†], **Xiaoyue Xu** [1], **Ying Yang** [1], **Jianli Wang** [2], **Dongmei Zhang** [2], **Linlin Mu** [2], **Xu Zhuang** [2], **Zhongbao Shen** [2,*] and **Changhong Guo** [1,*]

[1] Key Laboratory of Molecular and Cytogenetics, College of Life Science and Technology, Harbin Normal University, Harbin 150025, China; wz1528487057@163.com (Z.W.); 15645165258@163.com (X.X.); yangying1212@126.com (Y.Y.)

[2] Institute of Forage and Grassland Sciences, Heilongjiang Academy of Agricultural Sciences, Harbin 150086, China; rokiyou@126.com (J.Y.); wangjianlivip@126.com (J.W.); zhd_mei@163.com (D.Z.); 13030086918@163.com (X.Z.)

[*] Correspondence: shzhb1973@126.com (Z.S.); kaku3008@hrbnu.edu.cn (C.G.)

[†] These authors contributed equally to this work.

**Abstract:** Sweet clover (*Melilotus albus*) is a high-quality leguminous forage grass with salinity tolerance, drought tolerance, and cold hardiness. We selected four varieties of sweet clover with different sensitivities (061898, 061930, No. 55 white flower, and Ningxia white flower) and analyzed the effects of different concentrations of three sodium salts ($Na_2CO_3$, $NaHCO_3$, and $NaCl$) on their physiology and biochemistry responses. Growth and development indexes (such as germination rate, root length, shoot length), chlorophyll content, osmotic regulators (proline, soluble sugar), malondialdehyde (MDA), superoxide dismutase (SOD), peroxidase (POD), and catalase (CAT) were determined under saline–alkali stress. Seed germination and seedling growth of all four clover species were significantly inhibited under saline–alkali stress. During germination, seed germination rate, root length, and shoot length decreased with increasing saline and alkaline concentration. Under saline–alkali stress, chlorophyll content tended to increase and then decrease, cell damage and death increased, and malondialdehyde, soluble sugar, and proline content tended to increase and then decrease. Moreover, the activities of SOD, POD, and CAT all increased and then decreased. Under $Na_2CO_3$ stress, the decrease in chlorophyll content of the resistant variety 061898 was less than in the sensitive Ningxia white flower variety. As the concentrations of $Na_2CO_3$, $NaHCO_3$, and $NaCl$ increased, the maximum photochemical efficiency of PSII was significantly affected. The resistant 061898 is capable of maintaining higher photosynthetic efficiency. Furthermore, under treatments with the three kinds of saline–alkali solutions, cell damage and death for Ningxia white flower were greater than in 061898. For 061898, the increases in soluble sugar and proline content were greater and the increase in malondialdehyde content was less, while the antioxidant enzyme activities were higher than those in Ningxia white flower. All four sweet clover varieties had higher stress resistance with neutral than with alkaline salts. When stressed by medium to high saline–alkali concentrations, sweet clover seedlings had increased osmotic substance content, enhanced antioxidant enzyme activity, and regulated physiological metabolism. Additionally, sweet clovers regulated the expression of *WRKY33*, *GH3*, *CYCD3*, *OXI1*, *MKK2*, *MYC2*, *JAZ*, *COI1*, *PYL*, *PP2C*, *TGA*, and *MPK3* to adapt to the saline–alkali environment and improve saline–alkali tolerance. Our analysis of the sweet clover salinity tolerance mechanism contributes to its further use and is of significant importance for addressing land salinization and promoting sustainable agricultural and pastoral practices in China.

**Keywords:** saline–alkaline tolerance; *Melilotus albus*; stress response; gene expression analysis; crop utilization

## 1. Introduction

Soil salinization seriously affects agricultural production and restricts sustainable economic development. Now, it has become a topic of global concern, with a worldwide saline–alkali area of about $8.0 \times 10^9$ hm$^2$ [1], of which about $9.9 \times 10^7$ hm$^2$ is in China [2]. Saline–alkali soil is typically identifiable by salt content and pH level (above pH 8). Under natural conditions, salt stress is mainly caused by neutral salt (e.g., NaCl, Na$_2$SO$_4$), whereas saline–alkali stress is usually induced by basic salts (e.g., NaHCO$_3$, Na$_2$CO$_3$ [3,4]). Saline–alkali soil changes soil physicochemical properties and impedes plant nutrient uptake. Saline–alkali stress damage plants mainly through osmotic stress, ionic toxicity, nutrient imbalance, and oxidative stress [5]. For example, severely alkali-stressed plants will significantly accumulate MDA and ROS contents [6]. Under saline–alkali stress conditions, plants are impact by hindrance to seed germination, growth, and development, as well as flowering and fruiting [7,8]. Therefore, saline–alkali soil has become a major limiting factor for crop production in global agriculture [9].

Considering the severity of plant damage caused by soil salinization and alkalization, it is crucial to study the response mechanisms of plants under saline–alkali stress. To mitigate the damage and toxicity, plant have to develop various strategies to adapt to saline–alkali environments. These strategies include synthesis of osmoregulatory substances (such as proline, soluble sugar), enhancing the activities of antioxidant enzymes (such as SOD, POD, CAT), and upregulating saline–alkaline-responsive genes [10].

Sweet clover (*Melilotus albus*) is an annual or biennial herbaceous legume [11] and is cultivated worldwide [12]. Sweet clover has demonstrated remarkable environmental resilience that can be planted in saline–alkali pastures, where crops struggle to grow, because it possesses strong resistance to adversity, salt, and cold tolerance [13]. Widely used as fodder, green manure, and soil conservation crops [14], sweet clover has a higher yield and nitrogen-fixation ability relative to alfalfa [15]. As an excellent leguminous forage grass, sweet clover is rich in nutritional value, crude protein, and crude fat contents; it is also a valuable research subject for studying the salinity tolerance mechanism of high-quality forage grasses [16]. When planted in mildly saline–alkali soils, it can reduce soil salinity and enhance soil fertility, achieving soil improvement [17]. As one of the candidate plants for ecological restoration and reconstruction of salinized abandoned land, with a well-developed root system and tall stature, sweet clover can be used for wind and sand stabilization as well as soil and water conservation. Sweet clover produces infinite inflorescences over a long flowering period, making it a good nectar plant [18].

Studies on sweet clover's saline–alkali tolerance have been conducted by researchers. These results reveal that multiple gene families are involved in the response of sweet clover to salt stress [19,20]. However, research on the physiological and molecular mechanisms underlying sweet clover's adaptation to saline–alkali environments remains incomplete and superficial. We selected four sweet clover varieties with varying levels of saline–alkali tolerance to study their physiological characteristics under saline–alkali stress. We conducted experimental tests to measure MDA and proline concentrations, photosynthesis, reactive oxygen species accumulation, antioxidant enzymes activity, and stress-responsive gene expression in sweet clover subjected to saline–alkali conditions. The objectives of this research were to analyze the response mechanisms of sweet clover to saline–alkali stress and compare the differences in adaptation mechanisms to saline–alkali stress. The results of this study can provide theoretical support for the breeding of sweet clover and guide its cultivation in saline–alkali soils.

## 2. Materials and Methods

### 2.1. Plant Materials and Treatment

Seeds of four varieties of sweet clover, 061898, 061930, No. 55 white-flowered (55WF), and Ningxia white flowered (NXWF), were provided by the Pasture Breeding Research Laboratory, Grass Research Institute, Heilongjiang Provincial Academy of Agricultural Sciences.

Full, healthy, uniformly sized sweet clover seeds were selected. First, they were polished with sandpaper to break up the seed hardness, then treated with 75% ethanol for 30 s. Then the seeds were soaked in 5% NaClO for 5 min, rinsed with sterile water 3–5 times, and placed in square petri dishes with a diameter of 10 cm lined with 2 layers of filter paper. Twenty seeds were placed in each dish and $Na_2CO_3$ solutions (0, 10, 20, 30, 40, and 50 mmol/L), $NaHCO_3$ solutions (0, 30, 50, 70, and 90 mmol/L), and NaCl solutions (0, 50, 100, 150, 200, 250, and 300 mmol/L) were added to the dishes as stress treatments. Three replicates were performed for each treatment. Germination was carried out in an incubator (Ningbo Jiangnan Instrument Factory, Ningbo, China) at a constant temperature of 25 °C with a light period of 18 h/6 h (light/dark). This was appropriately supplemented with sterile water based on the rate of evaporation to maintain relatively stable concentrations. Sweet clover seed germination time is 7 d.

Sweet clover seeds were planted in pots with a diameter of 11 cm and depth of 11 cm filled with sterilized vermiculite, irrigated weekly with Hoagland nutrient solution, and placed in a light incubator (Ningbo Jiangnan Instrument Factory, Ningbo, China) with 16 h light/8 h dark and at 25/22 °C temperature. After 1 month of growth, they were stressed with one of 3 treatments: 0, 10, 20, 30, 40, and 50 mmol/L $Na_2CO_3$ solution; 0, 30, 50, 70, and 90 mmol/L $NaHCO_3$ solution; or 0, 50, 100, 150, 200, 250, and 300 mmol/L NaCl solution, respectively. Aboveground parts were collected after 10 d to determine physiological indices and gene expression level.

*2.2. Methods*

2.2.1. Seed Germination

Seed germination indexes were determined using the International Seed Inspection Protocol [21], including germination percentage, root length, and shoot length. Germination rate (%) = No. of germinated seeds on the seventh day/No. of seeds for testing × 100%; relative germination rate (%) = germination rate at each salt concentration/control germination rate × 100 percent; sweet clover root length and shoot length were determined on day 7 and repeated five times.

2.2.2. Physiological Analysis

For chlorophyll, 0.1 g fresh leaf was extracted in 95% ethanol till leaf discoloration and was tested spectrophotometrically at absorbances of 665 nm and 649 nm [22]. Three replicates were designed for each condition. Chlorophyll fluorescence imaging of the whole plant was determined using the Multi-functional Photosynthetic Phenotypic Measurement System for Plants (PlantExplorer Pro+, Pheno Vation, Wageningen, The Netherlands).

Leaves after 10 days of saline–alkali stress induced WT were incubated in di-amino-benzine (DAB, 1 mg/mL) or nitroblue tetrazolium (NBT, 0.1 mg/mL) solution overnight at room temperature in the dark to detect the accumulation of $H_2O_2$ and $O_2^-$, respectively. For Evan's blue staining, excised leaves were incubated in 0.1% Evan's blue solution overnight. The stained leaves were soaked in 90% ethanol to remove the chlorophyll and subsequently photographed. All the experiments were performed before and after stress treatments mentioned in respective figures [23,24].

The proline content was measured using acid ninhydrin colorimetry using pure proline as a standard. The soluble sugar was determined by anthrone colorimetry using glucose as a standard. For their measurements, about 0.5 g of fresh seedling was used with three replicates. The absorbance was recorded at wavelengths of 520 nm and 625 nm for proline and soluble sugar, respectively, on a UV spectrophotometer [25,26].

MDA level was evaluated by the thiobarbituric acid (TBA) method. About 0.5 g of fresh seedling was homogenized in 5 mL of 10% trichloroacetic acid (TCA) solution for further experiments with three replicates. The absorbance of the last supernatant was recorded at wavelengths of 450, 532, and 600 nm, respectively [27].

In order to assess the activities of SOD, POD, and CAT in the seedling, about 0.5 g of the fresh sample was weighed with three replicates. These enzymatic antioxidant

compounds (SOD, POD, and CAT) were extracted and tested with UV spectrophotometer at absorbances of 560 nm, 470 nm, and 240 nm, respectively [28,29].

### 2.2.3. Physiological Analysis

The tissues of seedlings were harvested at 10th day after salt and alkali treatment and under normal growth conditions with three independent biological replicates per treatment. TRIzol reagent (Invitrogen, Carlsbad, CA, USA) was used to extract RNA according to the manufacturer's instructions. RNA samples were reverse-transcribed using the TransScript One-Step gDNA Removal and cDNA Synthesis SuperMix Kit (Transgen, Beijing, China). Primers were designed by Primer 5.0.

qRT-PCR reaction was performed with Roche Lightcycler 96 using ChamQ Universal SYBR qPCR Master Mix (Vazyme, Nanjing, China) following the manufacturer's instructions. The cycle conditions for qRT-PCR were as follows: 95 °C for 30 s, followed by 40 cycles of 95 °C for 10 s and 60 °C for 30 s. Actin was used as an internal control, and the relative transcript levels were analyzed based on the $2^{-(\Delta\Delta Ct)}$ method.

### 2.2.4. Statistical Analysis

Experimental results are expressed as mean $\pm$ standard error (SE). SPSS 22.0 statistical software was used for data analysis and Origin2022 for graphing. Data were tested for significance by one-way analysis of variance (One-Way ANOVA) at the 95% level. The Duncan post hoc test was used to test for significant differences in multiple comparisons between treatments ($p < 0.05$).

## 3. Results

### 3.1. Effect of Saline–Alkali Stress on Sweet Clover Germination

#### 3.1.1. Effect of Saline–Alkali Stress on Sweet Clover Germination Rate

The germination rates of all four sweet clover varieties gradually decreased with increasing saline–alkali stress concentration, but the degree of germination inhibition between varieties differed (Figure 1). The germination rate of 061898 at 20 mmol/L $Na_2CO_3$ treatment was not significantly different ($p > 0.05$) from that of the control (CK), and at 30 mmol/L $Na_2CO_3$ treatment, it still reached 62%, which was significantly higher than the other three varieties. The germination rate of NXWF at 20 mmol/L $Na_2CO_3$ treatment was significantly different from that of the control (CK), and it did not germinate under 30 mmol/L $Na_2CO_3$ treatment (Figure 1A). At 50 mmol/L $NaHCO_3$ treatment, 061898 and 061930 germination rates were 58% and 50%, respectively, showing good tolerance, while those of 55WF and NXWF decreased sharply and significantly compared to the control (CK) with weaker tolerance (Figure 1B). At an NaCl solution concentration of 250 mmol/L, 061898 germination rate still reached 17%, which was significantly higher than those of the other three sweet clover germplasm seeds (Figure 1C).

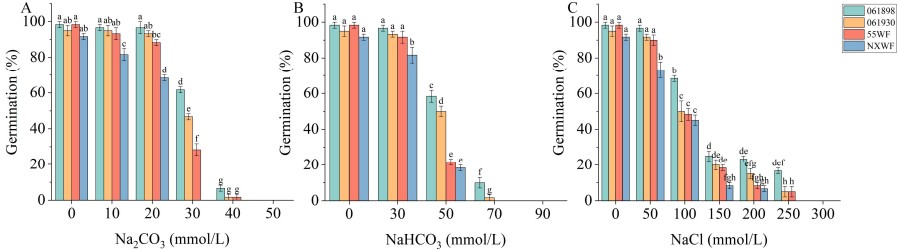

**Figure 1.** Effect of saline–alkali stresses on the germination of sweet clover seeds. The germination under $Na_2CO_3$ (**A**); $NaHCO_3$ (**B**); NaCl (**C**) treatments. Different small letters above the columns indicate significant ($p < 0.05$) differences between different concentrations. The type 061898 is represented by green, 061930 is represented by orange, 55WF is represented by red, and NXWF is represented by blue. Values given are the mean $\pm$ SE ($n = 3$). Data were tested for significance by one-way ANOVA at $p < 0.05$.

The effects of $Na_2CO_3$, $NaHCO_3$, and NaCl solutions on 061898 germination rate were not significantly different from that of the control (CK) at low stress concentrations, but the inhibitory effect on germination rate gradually increased with increasing saline–alkali concentration. When the concentration of all three saline–alkali solutions reached 50 mmol/L, 061898 germination rate under NaCl treatment was the highest and under $Na_2CO_3$ treatment was zero.

When the $Na_2CO_3$ and $NaHCO_3$ solution concentration reached 50 mmol/L, 061930 germination rate was significantly different from that of the control (CK) treatment, but there was no significant difference between the germination rate of 061930 and the control at a 50 mmol/L NaCl solution.

The tolerance trend to the $Na_2CO_3$, $NaHCO_3$, and NaCl saline–alkali solution stresses of 55WF and NXWF was the same in the first two sweet clover varieties, but the germination rate was 0 when the $Na_2CO_3$ and $NaHCO_3$ solution was 50 mmol/L.

### 3.1.2. Effect of Saline–Alkali Stress on Sweet Clover Root Length

Under different saline–alkali stresses, root length of all four sweet clover species shortened with increasing saline–alkali concentration, but there were differences in the salt concentration at which the root length was completely inhibited (Figure 2). When the $Na_2CO_3$ solution reached 30 mmol/L, the root length of 061898 was shorter than that of 061930, which was equal to 55WF, while NXWF was completely suppressed (Figure 2A). The $NaHCO_3$ solution stress effect on the root length of 061930 and NXWF was low and lower than 061898 and 55WF when at 30 mmol/L (Figure 2B). Although the effect of NaCl solution stress on white flower No. 55 root length growth was lower than the other three varieties, it was greater when the solution concentration was 200 mmol/L (Figure 2C).

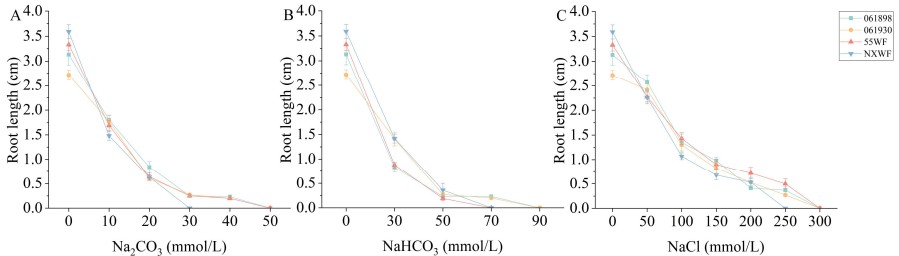

**Figure 2.** Effect of saline–alkali stresses on root length of sweet clover. The root length under $Na_2CO_3$ (**A**); $NaHCO_3$ (**B**); NaCl (**C**) treatments. The type 061898 is labeled with a green square, 061930 labeled with an orange dot, 55WF labeled with a red equilateral triangle, and NXWF labeled with a blue inverted triangle. Values represent mean $\pm$ SE ($n = 5$). Data were tested for significance by one-way ANOVA at $p < 0.05$.

$Na_2CO_3$ and $NaHCO_3$ solutions had a strong effect on root length, and when the concentration reached 30 mmol/L, the root length was shortened to <1 cm, which differed significantly from that of the control (CK). The root length of 061898 was also shortened to <1 cm but only when the concentration of NaCl solution reached 150 mmol/L. All four varieties exhibited the same trend under $Na_2CO_3$, $NaHCO_3$, and NaCl solution stresses, respectively, showing more tolerance to NaCl stress.

### 3.1.3. Effect of Saline–Alkali Stress on Sweet Clover Shoot Length

The shoot length declined with increasing $Na_2CO_3$, $NaHCO_3$, and NaCl solution concentrations, but for different sweet clover varieties, it varied slightly (Figure 3). Under $Na_2CO_3$ stress, 061898 shoot length was consistently greater than those of the other three varieties, and when it reached 40 mmol/L, it remained at 0.43 cm (Figure 3A). When the $NaHCO_3$ solution was 50 mmol/L, the bud length was reduced to 0.43 cm in 55WF, while in the other three varieties, it was above 0.5 cm (Figure 3B). When the NaCl solution was 200 mmol/L, shoot lengths of the four varieties were ranked in the following order: 55WF > NXWF > 061930 > 061898 (Figure 3C).

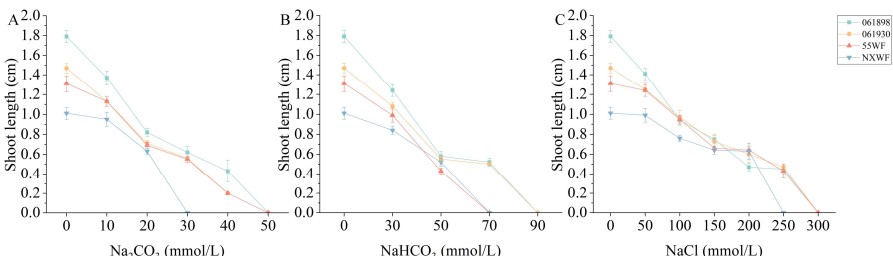

**Figure 3.** Effect of saline–alkali stresses on shoot length of sweet clover. The shoot length under Na$_2$CO$_3$ (**A**); NaHCO$_3$ (**B**); NaCl (**C**) treatments. The type 061898 is labeled with a green square, 061930 labeled with an orange dot, 55WF labeled with a red equilateral triangle, and NXWF labeled with a blue inverted triangle. Values represent mean ± SE (*n* = 5). Data were tested for significance by one-way ANOVA at *p* < 0.05.

The effects of Na$_2$CO$_3$ and NaHCO$_3$ on 061898 shoot length were greater than NaCl. When Na$_2$CO$_3$ and NaHCO$_3$ solutions reached 20 and 50 mmol/L, respectively, 061898 shoot length had been shortened to <1.0 cm, which was significantly lower than the control (CK), and at 100 mmol/L, the shoot length was 0.94 cm. The same trend was observed in the shoot lengths of 061930, 55WF, and NXWF, under the stress of Na$_2$CO$_3$, NaHCO$_3$, and NaCl solutions, respectively.

### 3.2. Effect of Saline–Alkali Stress on Sweet Clover Seedlings

3.2.1. Effect of Saline–Alkali Stress on Sweet Clover Chlorophyll Content

As the concentration of saline–alkali solutions increased, the chlorophyll content in the four varieties of sweet clover initially increased and then decreased (Figure 4). Under Na$_2$CO$_3$ solution stress, 061898 chlorophyll content increased significantly at 10 mmol/L, while the chlorophyll content of all four varieties decreased significantly when the solution concentration reached 50 mmol/L. Under Na$_2$CO$_3$ solution stress, the chlorophyll contents of 061898, 061930, 55WF, and NXWF decreased by 39%, 58%, 55% and 55%, respectively, compared with that of the control (Figure 4A). With NaHCO$_3$ solution stress, both 061898 and 061930 chlorophyll content increased slightly compared to that of the control treatment, but significantly dropped (43%, 40%, 39%, and 40%) at 90 mmol/L compared to the control (Figure 4B). Under low NaCl solution stress, the chlorophyll content of all four varieties was low and did not significantly differ from the control; however, when the solution concentration reached 300 mmol/L, the chlorophyll content declined by 42%, 46%, 41%, and 34%, respectively, compared with the control (Figure 4C).

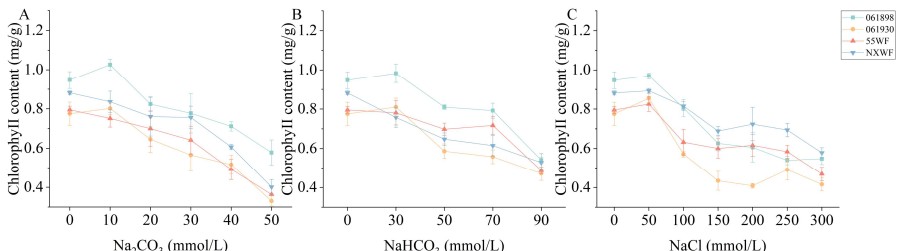

**Figure 4.** Effect of saline–alkali stress on chlorophyll content in sweet clover seedlings. The chlorophyll content under Na$_2$CO$_3$ (**A**); NaHCO$_3$ (**B**); NaCl (**C**) treatments. The type 061898 is labeled with a green square, 061930 labeled with an orange dot, 55WF labeled with a red equilateral triangle, and NXWF labeled with a blue inverted triangle. Values represent mean ± SE (*n* = 3). Data were tested for significance by one-way ANOVA at *p* < 0.05.

The chlorophyll content of sweet clover 061898 reached a maximum at Na$_2$CO$_3$ of 10 mmol/L, but under NaHCO$_3$ and NaCl solution stress, it increased at 30 mmol/L and 50 mmol/L, respectively, followed by a continuous decrease. The chlorophyll content of 061930 generally increased and then decreased under the three saline–alkali stresses, but

decreased more in $Na_2CO_3$ than in $NaHCO_3$ and NaCl. For 55WF, the chlorophyll content decreased as the solution concentration increased, reaching its minimum at concentrations of 50 mmol/L for $Na_2CO_3$, 90 mmol/L for $NaHCO_3$, and 300 mmol/L for NaCl, with each being significantly different from the control.

### 3.2.2. Effect of Saline–Alkali Stress on Sweet Clover Maximum Photochemical Quantum Yield of PSII

The maximum photochemical quantum yield (Fv/Fm) of photosystem II (PSII) reflects the efficiency of endowment light energy conversion in the PSII reaction center. A decreasing trend in this parameter was observed under stress conditions. Under $Na_2CO_3$ solution stress, the Fv/Fm difference was >0 after stress compared with beforehand in 061898, 061930, and 55WF, while the difference was negative in NXWF (Figure 5A). When the $NaHCO_3$ solution concentration reached 90 mmol/L, the difference in Fv/Fm was >0 after stress compared with beforehand in 061898 and 061930, whereas the difference was negative in 55WF and NXWF (Figure 5B). At NaCl solution concentrations up to 300 mmol/L, the Fv/Fm difference was >0 for all four varieties after stress compared with beforehand (Figure 5C).

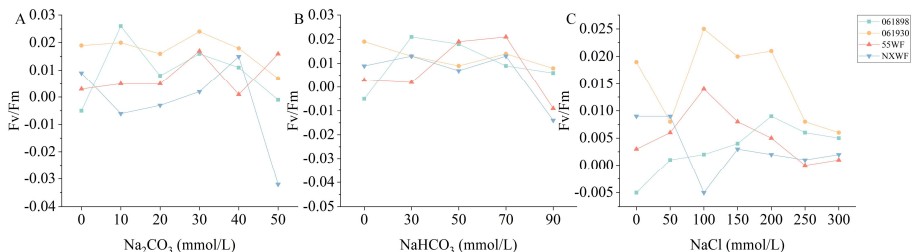

**Figure 5.** Effect of saline–alkali stresses on Fv/Fm of sweet clover. The Fv/Fm under $Na_2CO_3$ (**A**); $NaHCO_3$ (**B**); NaCl (**C**) treatments. The type 061898 is labeled with a green square, 061930 labeled with an orange dot, 55WF labeled with a red equilateral triangle, and NXWF labeled with a blue inverted triangle.

At $Na_2CO_3$, $NaHCO_3$, and NaCl solution concentrations of 50, 90, and 300 mmol/L, respectively, the difference between post-stress and pre-stress Fv/Fm values were negative, positive, and positive, respectively, and the recovery ability of leaves under $NaHCO_3$ and NaCl solution stresses was higher than that of 061898 $Na_2CO_3$ stress-treated leaves. The same trend was observed in the other three varieties, all of which exhibited better recovery ability at the NaCl stress-treated seedling stage.

Under low concentration stress of the three saline–alkali solutions, 061898 did not show significant damage and was mostly green, but when the concentration of the three saline–alkali solutions reached its maximum, slight reddish-yellow damage was observed. Similar trends were observed in the white flowers of 061930 and 55WF, and the damage was greatest in the former. NXWF was more seriously damaged when the concentration of the three saline–alkali solutions was at maximum, with more leaves reddish-yellow and with $Na_2CO_3$ causing more serious damage to the photosynthetic system (Figure 6).

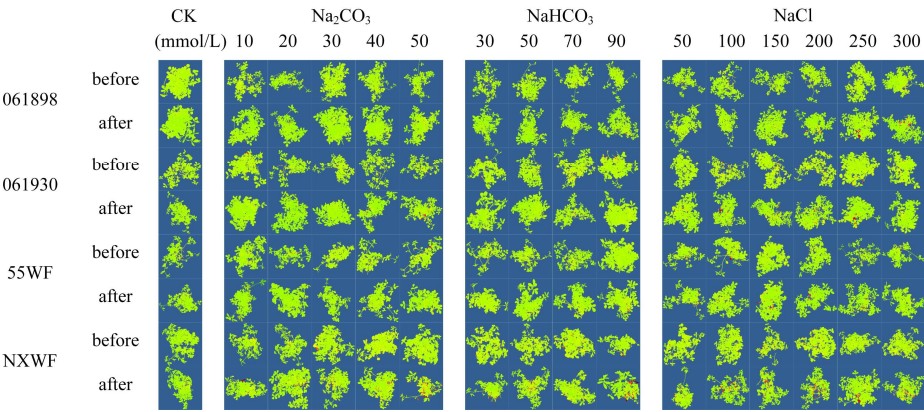

**Figure 6.** Fluorescence imaging of Fv/Fm of in sweet clover under saline–alkali stress. The severity of damage is represented by the progression from weak to strong fluorescence, indicated by green, yellow, and red, respectively. Fluorescence observation through Multi-functional Photosynthetic Phenotypic Measurement System for Plants.

### 3.2.3. Effects of Saline–Alkali Stress on Sweet Clover ROS Accumulation and Cellular Damage in Leaves

To gauge the extent of sweet clover cellular damage under saline–alkali stress conditions, the accumulation of two important ROS elements ($H_2O_2$ and $O_2^-$) was measured in leaf cells of the four varieties of sweet clover by using DAB and NBT staining, respectively. In the control, there was little leaf accumulation of $H_2O_2$ and $O_2^-$ in the four varieties of sweet clover (Figures 7 and 8), while $H_2O_2$ and $O_2^-$ gradually accumulated with increasing concentrations of the three salts. Under the three saline–alkali stress conditions, 061898 accumulated the lowest levels of $H_2O_2$ and $O_2^-$, while NXWF had higher leaf levels after the stress treatments, and dark brown (DAB) and dark blue (NBT) staining was detected.

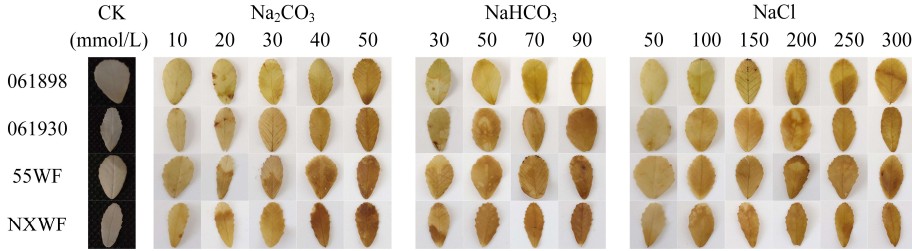

**Figure 7.** Analysis of $H_2O_2$ accumulation caused by saline–alkali stress in sweet clover leaves. Detection of $H_2O_2$ accumulation by di-aminobenzine (DAB) staining. For the histochemical assays, stained leaves were soaked in 90% ethanol to remove the chlorophyll and then photographed.

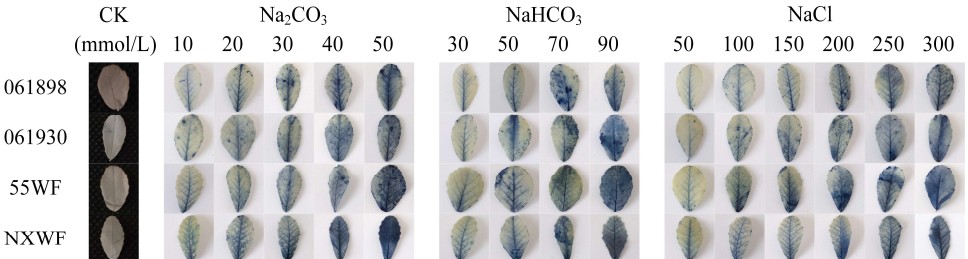

**Figure 8.** Analysis of $O_2^-$ accumulation caused by saline–alkali stress in sweet clover leaves. Detection of $O_2^-$ accumulation by nitroblue tetrazolium (NBT) staining. For the histochemical assays, stained leaves were soaked in 90% ethanol to remove the chlorophyll and then photographed.

### 3.2.4. Effect of Saline–Alkali Stress on Sweet Clover Leaf Cell Death

The detection of plant cell activity is a direct and effective method to determine the degree of plant injury. In this experiment, Evans blue staining was used to detect cell death in sweet clover leaves after different salt treatments. Leaves of the four varieties were barely stained blue in the controls, but as the saline–alkali solution concentration increased, the blue-stained area gradually enlarged, the degree of staining deepened, and the number of dead cells increased. When the $Na_2CO_3$, $NaHCO_3$, and $NaCl$ solutions reached 50, 90, and 300 mmol/L, respectively, the stained area was the smallest in 061898 and the largest in NXWF, and cell death in 061898 was significantly lower than in NXWF (Figure 9).

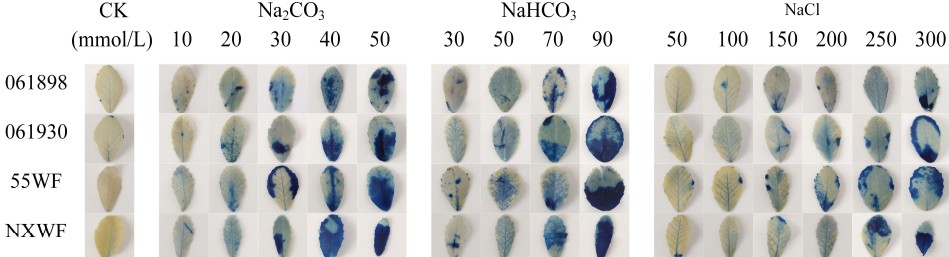

**Figure 9.** Analysis of cell death caused by saline–alkali stress in sweet clover leaves. Detection of cell death by Evan's blue staining. For the histochemical assays, stained leaves were soaked in 90% ethanol to remove the chlorophyll and then photographed.

### 3.2.5. Effect of Saline–Alkali Stress on Sweet Clover Soluble Sugar Content

The types 061898, 061930 and 55WF had peak soluble sugar contents at an $Na_2CO_3$ solution of 30 mmol/L, which were 37%, 32%, and 30%, respectively, higher than that of the control (Figure 10). The peak content of NXWF occurred at a concentration of 20 mmol/L, which was 27% greater than that of the control. When the $Na_2CO_3$ solution was 50 mmol/L, NXWF soluble sugar content decreased by 4% compared with that of the control, but in the remaining three varieties, it increased slightly (Figure 10A). At the maximum $NaHCO_3$ solution concentration, the soluble sugar content of the four varieties increased by 34%, 27%, 26%, and 12%, respectively, compared to that of the control (Figure 10B). At an $NaCl$ solution of 200 mmol/L, the soluble sugar content of all four varieties increased significantly compared to that of the control, and at 300 mmol/L, the soluble sugar content increased by 31%, 25%, 25%, and 20%, respectively (Figure 10C). Overall, the four sweet clover varieties showed better tolerance to neutral salts than to alkaline salts when the same concentration of three different saline–alkali solutions were reached, with a greater tolerance to $NaHCO_3$ than to $Na_2CO_3$.

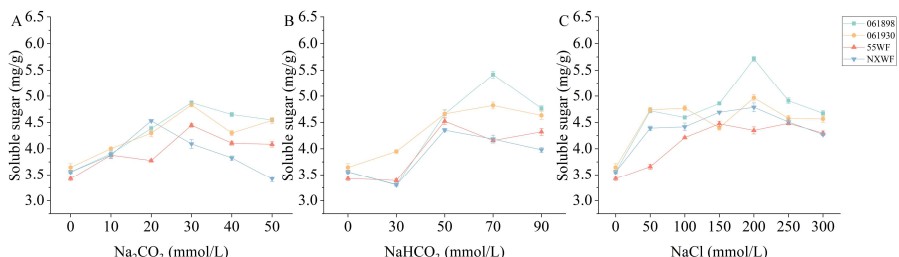

**Figure 10.** Effect of saline–alkali stress on soluble sugar content in sweet clover seedlings. The soluble sugar content under $Na_2CO_3$ (**A**); $NaHCO_3$ (**B**); $NaCl$ (**C**) treatments. The type 061898 is labeled with a green square, 061930 labeled with an orange dot, 55WF labeled with a red equilateral triangle, and NXWF labeled with a blue inverted triangle. Values represent mean ± SE (*n* = 3). Data were tested for significance by one-way ANOVA at *p* < 0.05.

### 3.2.6. Effect of Saline–Alkali Stress on Sweet Clover Proline Content

Generally, the proline content of the four varieties tended to increase with increasing saline–alkali solution concentrations (Figure 11). At an $Na_2CO_3$ solution of 50 mmol/L,

the proline content in all four varieties showed increases of 67%, 46%, 42%, and 26%, respectively, compared to the control (Figure 11A). A same trend was observed at an NaHCO$_3$ solution of 90 mmol/L, which increased by 89%, 74%, 63%, and 58%, respectively, compared to the control, with 061898 increasing significantly more than NXWF (Figure 11B). Under NaCl stress conditions, the proline content initially increased and then decreased, with varying decreases at solution concentrations from 250 to 300 mmol/L. However, the proline content was always higher than in the control, increasing by 137%, 77%, 100%, and 96%, respectively, greater than the control at 300 mmol/L (Figure 11C). The four varieties were less tolerant to Na$_2$CO$_3$ at the same saline–alkali solution concentrations.

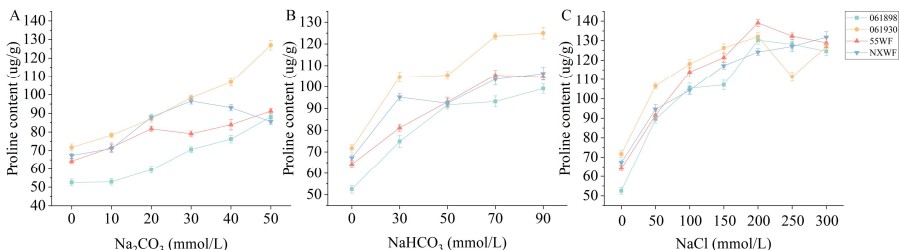

**Figure 11.** Effect of saline–alkali stress on proline content in sweet clover seedlings. The proline content under Na$_2$CO$_3$ (**A**); NaHCO$_3$ (**B**); NaCl (**C**) treatments. The type 061898 is labeled with a green square, 061930 labeled with an orange dot, 55WF labeled with a red equilateral triangle, and NXWF labeled with a blue inverted triangle. Values represent mean $\pm$ SE (*n* = 3). Data were tested for significance by one-way ANOVA at *p* < 0.05.

### 3.2.7. Effect of Saline–Alkali Stress on Sweet Clover Malondialdehyde Content

Under saline–alkali stress conditions, intracellular oxidative damage and membrane structure damage occur, and MDA content rises, which reduces cell membrane stability and exacerbates membrane system damage and cellular senescence. Generally, the leaf MDA content decreased and then increased in the four sweet clover varieties as saline–alkali solution concentrations increased (Figure 12). When the Na$_2$CO$_3$ solution reached 50 mmol/L, MDA contents of the four varieties were all significantly higher than that of the control, by 111, 154, 167, and 219%, respectively, with the rise in NXWF significantly higher than in 061898 (Figure 12A). At an NaHCO$_3$ solution of 90 mmol/L, the MDA content of the four varieties reached the maximum for each treatment concentration, with increases of 108, 128, 170, and 171%, respectively, compared to that of the control (Figure 12B). Under NaCl solution stress, the MDA contents of the four varieties were 92%, 126%, 143%, and 145%, respectively, higher than the control at a concentration of 300 mmol/L (Figure 12C).

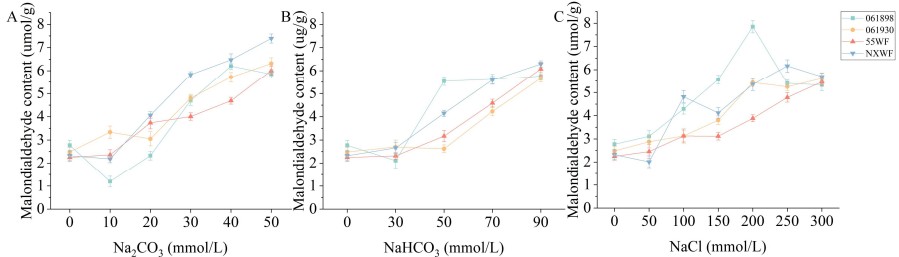

**Figure 12.** Effect of saline–alkali stress on MDA content in sweet clover seedlings. The MDA content under Na$_2$CO$_3$ (**A**); NaHCO$_3$ (**B**); NaCl (**C**) treatments. The type 061898 is labeled with a green square, 061930 labeled with an orange dot, 55WF labeled with a red equilateral triangle, and NXWF labeled with a blue inverted triangle. Values represent mean $\pm$ SE (*n* = 3). Data were tested for significance by one-way ANOVA at *p* < 0.05.

The MDA content of 061898 decreased and was lower than that of the control when Na$_2$CO$_3$ and NaHCO$_3$ solutions were 10 and 30 mmol/L, respectively. It then began to increase and reached a maximum when Na$_2$CO$_3$, NaHCO$_3$, and NaCl solutions reached 40,

90, and 200 mmol/L, respectively. Varieties 061930, 55WF, and NXWF exhibited a similar trend in MDA content changes. The four varieties experienced increasing stress in the following order: $Na_2CO_3$ > $NaHCO_3$ > NaCl.

### 3.2.8. Effect of Saline–Alkali Stress on Sweet Clover SOD Activity

Generally, among the four sweet clover varieties, the SOD activity increased and then decreased with increasing saline–alkali solution concentrations (Figure 13). At an $Na_2CO_3$ solution concentration of 10 mmol/L, the varieties exhibited minimal change except for NXWF, which demonstrated a decrease in SOD activity compared to that in the control. At the maximum concentration, all four varieties showed increases in SOD activity compared to that of the control, with the respective increases being 59%, 38%, 38%, and 23% (Figure 13A). The order of increase in SOD activity of the four varieties under $NaHCO_3$ solution stress was 061898 > 061930 > 55WF > NXWF (Figure 13B). Under NaCl stress conditions, NXWF's SOD activity peaked at 150 mmol/L, which was 44% higher than that of the control, and the four varieties increased in the same order as $NaHCO_3$ at 300 mmol/L (Figure 13C). All four varieties showed a smaller increase in SOD under stress with $Na_2CO_3$ solution than with $NaHCO_3$ and NaCl when stressed with high saline–alkali solution concentrations.

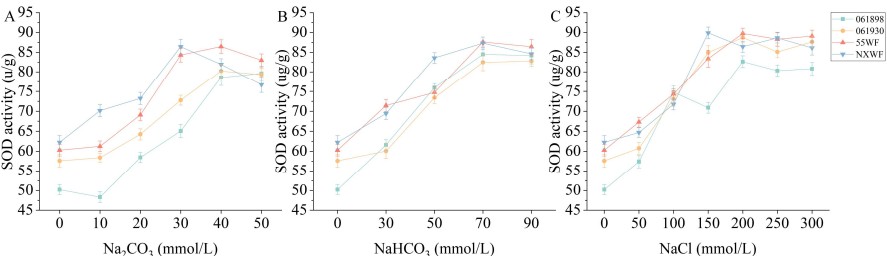

**Figure 13.** Effect of saline–alkali stress on SOD activity in sweet clover seeding. The SOD activity under $Na_2CO_3$ (**A**); $NaHCO_3$ (**B**); NaCl (**C**) treatments. The type 061898 is labeled with a green square, 061930 labeled with an orange dot, 55WF labeled with a red equilateral triangle, and NXWF labeled with a blue inverted triangle. Values represent mean $\pm$ SE ($n$ = 3). Data were tested for significance by one-way ANOVA at $p < 0.05$.

### 3.2.9. Effect of Saline–Alkali Stress on Sweet Clover POD Activity

At lower saline–alkali stress concentrations, the leaf POD activity of the four sweet clover varieties increased gradually with increasing solution concentrations, but when the saline–alkali solution concentration increased to a certain value, it began to decline (Figure 14). The $Na_2CO_3$ concentration at 30–40 mmol/L increased the leaf POD activity of the four varieties to a maximum, which differed significantly from that of the control. At 50 mmol/L, the POD activity was 13%, 12%, 14%, and 5% higher for the four varieties than the control (Figure 14A). At an $NaHCO_3$ solution of 50 mmol/L, 55WF had the highest POD activity, and the remaining three varieties showed a peak at 70 mmol/L and then began to decline. By 90 mmol/L, the POD activity increased by 23%, 23%, 19%, and 12%, respectively, compared with that of the control (Figure 14B). The same trend was observed under NaCl stress conditions, with a decrease in POD activity appearing after the concentration was increased to 150–200 mmol/L (Figure 14C). For the three saline–alkali solutions at high concentrations, all four sweet clover varieties showed a greater POD activity increase in neutral, rather than in alkaline salts, with $NaHCO_3$ being greater than $Na_2CO_3$.

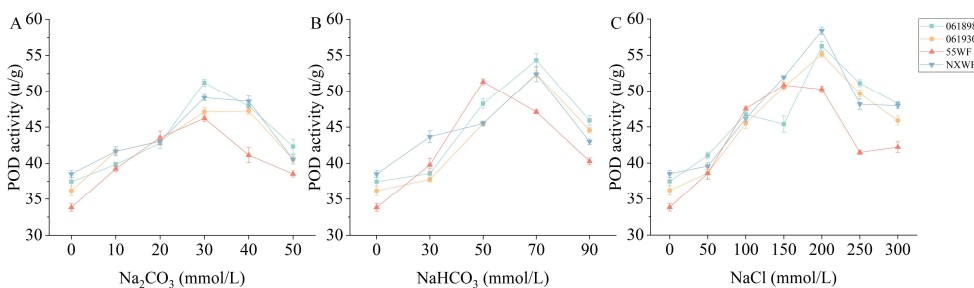

**Figure 14.** Effect of saline–alkali stress on POD activity in sweet clover seeding. The POD activity under $Na_2CO_3$ (**A**); $NaHCO_3$ (**B**); NaCl (**C**) treatments. The type 061898 is labeled with a green square, 061930 labeled with an orange dot, 55WF labeled with a red equilateral triangle, and NXWF labeled with a blue inverted triangle. Values represent mean ± SE ($n$ = 3). Data were tested for significance by one-way ANOVA at $p < 0.05$.

### 3.2.10. Effect of Saline–Alkali Stress on Sweet Clover CAT Activity

As the three saline–alkali solution concentrations increased, the CAT activity increased. After reaching a certain concentration, the CAT activity began to decrease, but was always higher than that in the control treatment (Figure 15). The $Na_2CO_3$ solution concentration at 40 mmol/L had the maximum CAT activity for 55WF, while the remaining three varieties peaked at 30 mmol/L and then began to decline. At a 50 mml/L solution, the CAT activity increased by 22%, 20%, 15% and 12%, respectively, compared to that of the control (Figure 15A). With a 90 mmol/L $NaHCO_3$ solution, the CAT activity of the four varieties was 17%, 14%, 16%, and 14% higher, respectively, than those of the control (Figure 15B). The NaCl solution of 300 mmol/L showed an increase in all four varieties compared to the control, but the difference was not significant (Figure 15C). The CAT activity under alkaline salt stress conditions was greater than that under neutral salt.

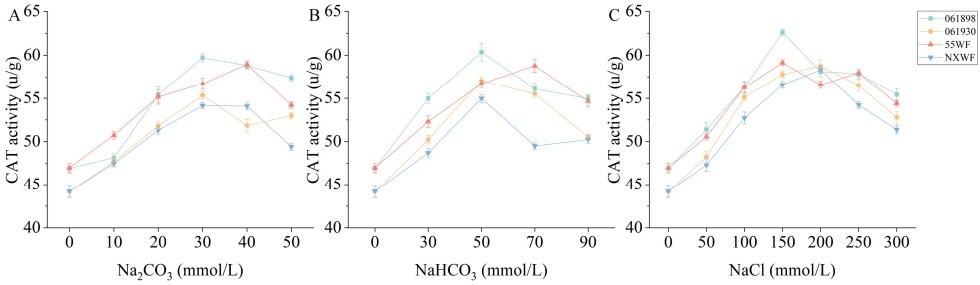

**Figure 15.** Effects of saline–alkali stress on CAT activity in sweet clover seeding. The CAT activity under $Na_2CO_3$ (**A**); $NaHCO_3$ (**B**); NaCl (**C**) treatments. The type 061898 is labeled with a green square, 061930 labeled with an orange dot, 55WF labeled with a red equilateral triangle, and NXWF labeled with a blue inverted triangle. Values represent mean ± SE ($n$ = 3). Data were tested for significance by one-way ANOVA at $p < 0.05$.

### 3.2.11. Gene Expression Level Analysis

Plants respond to saline–alkali stress by regulating the expression of various genes. We conducted an experiment to assess gene expression changes related to hormone signaling pathways and the MAPK pathway in salt-tolerant cultivar 061898 and salt-sensitive cultivar NXWF following treatment with 300 mmol/L NaCl, 50 mmol/L $Na_2CO_3$, and 90 mmol/L $NaHCO_3$ (Figure 16). After the neutral salt (NaCl) treatment, there were no changes in the expression levels of WRKY33, MYC2, JAZ, COI1, and PP2C in both 061898 and NXWF. CYCD3 was downregulated, while GH3, MKK2, PYL, and MPK3 were downregulated in 061898, except for OXI1, which was upregulated. In NXWF, GH3 was upregulated and TGA was downregulated following NaCl treatment, though these changes were not statistically different from those in the control.

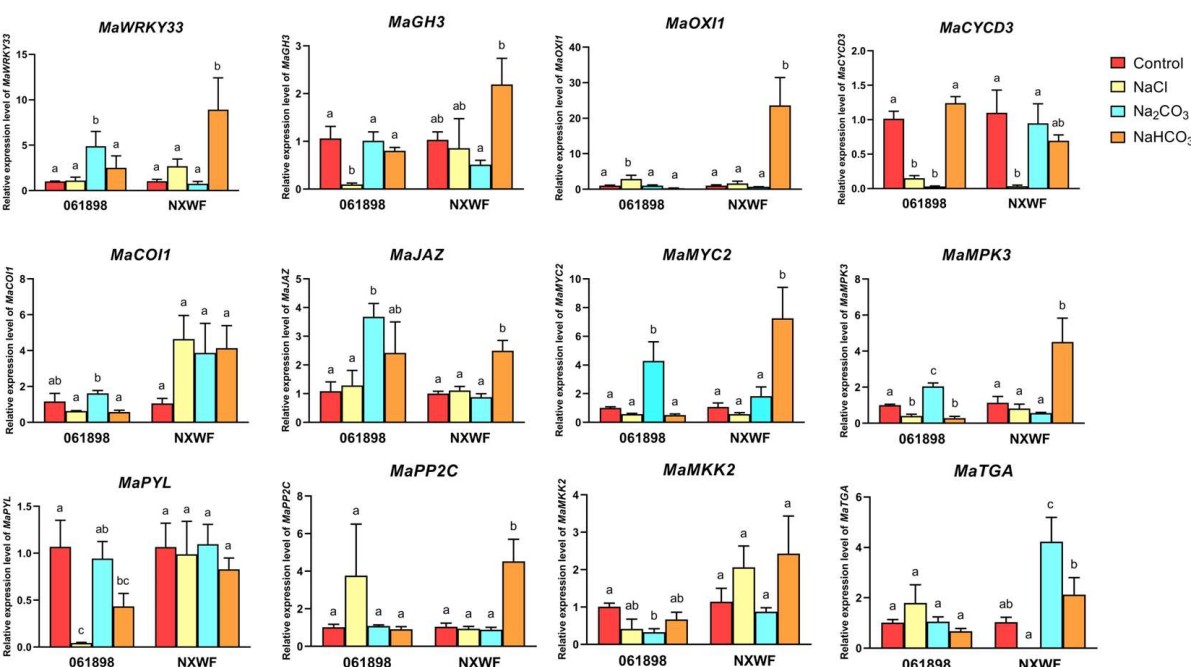

**Figure 16.** Gene expression level of 061898 and NXWF after saline–alkali stress. Different small letters above the columns indicate significant ($p < 0.05$) differences between different concentrations. The control is represented by red, NaCl is represented by yellow, $NaHCO_3$ is represented by blue, and $Na_2CO_3$ is represented by orange. Values given are the mean $\pm$ SE ($n = 3$).

Under $Na_2CO_3$ and $NaHCO_3$ alkaline stress, the deviations in gene expression patterns were greater than with NaCl stress. With mild alkaline stress, 061898 showed downregulation of PYL, MPK3, MKK2, and COI1; however, the changes in MKK2 and COI1 were not significant, and JAZ was upregulated but not significantly (Figure 16). In NXWF, the gene changes were more pronounced relative to 061898 (Figure 16), with significant upregulation of WRKY33, GH3, OXI1, MYC2, JAZ, PP2C, and MPK3. TGA was upregulated compared to the control, though not significantly, and CYCD3 was downregulated but not significantly. For severe alkaline stress, the changes in gene expression in 061898 were more apparent than NXWF, with significant upregulation of WRKY33, MYC2, JAZ, and MPK3 as well as significant downregulation of CYCD3 and MKK2. COI1 and PYL were upregulated and downregulated, respectively, but not significantly. For the sensitive material NXWF, following severe alkaline stress, only TGA showed a significant upregulation, while GH3 expression was downregulated but not significantly.

The findings indicate that sweet clover possesses a certain degree of saline–alkali tolerance, with more pronounced effects on gene expression from alkaline salts than neutral salts. Resistant and sensitive varieties may activate different genes in response to identical stress treatment, and the same variety may enact different molecular regulatory mechanisms for different saline–alkali treatments.

## 4. Discussion

### 4.1. Effect of Saline–Alkali Stress on Sweet Clover Germination

Tolerance to saline–alkali stresses varies among growth stages, with the germination stage being more sensitive to external stresses than other stages [30,31]. In this study, we observed that the germination rate was unaffected when seeds were subjected to low concentrations of saline–alkali treatment. However, high saline–alkali stress more strongly inhibited seed germination, which may be due to the reduction of seed osmoregulatory capacity by saline–alkali stress, thereby affecting their germination [32]. Additionally, saline–alkali stress disrupts seed cell membrane structure, which leads to metabolic disorders and reduced seed germination [33]. Studies have shown that the germination rate of

salinity-tolerant materials are generally higher than those of the sensitive varieties [34]. In this experiment, the germination rate of 061898 was still above 50% at an $Na_2CO_3$ concentration of 30 mmol/L, whereas NXWF seeds did not germinate. In our study, seeds of the four sweet clover varieties were most tolerant of NaCl, followed by $NaHCO_3$ and finally $Na_2CO_3$. Yanmin Lu et al. [35] also found that alkaline salts inhibit seed germination in *Trifolium repens* more than neutral salts.

A salt stress environment destroys the cell structure and inhibits normal shoot and root growth [36], in which the root system is the first part to be exposed to the saline–alkali environment and subjected to saline–alkali stress, which is mainly manifested in the osmotic potential of the water uptake being blocked [37], and the ionic toxicity and membrane permeability increased. We found that with increasing concentration, the three salts had a significant inhibitory effect on the embryonic root and germ of the four sweet clover varieties. Meanwhile, the inhibitory effect on the radicle was stronger than on the germ, which is consistent with the results of Liu Binshuo et al. [38] on the effect of seed germination of *Leymus chinensis* under saline–alkali stress.

### 4.2. Effect of Saline–Alkali Stress on Sweet Clover Seedlings

4.2.1. Effect of Saline–Alkali Stress on Chlorophyll Content and Chlorophyll Fluorescence

Conducting photosynthesis is an essential physiological process for plant life activities, and chlorophyll is the basis for photosynthesis [39]. Under normal growth conditions, plant chlorophyll content exhibits dynamic equilibrium, but saline–alkali stress disrupts this balance, causing changes in the chlorophyll content [40]. The chlorophyll content changes can reflect stress factors on the plant and measure the strength of the plant's saline–alkali tolerance [41]. In our experiment, the chlorophyll content of the four sweet clover varieties initially increased and then decreased. Under low saline–alkali concentration stress, the leaf chlorophyll content was elevated higher than that in the control treatment, indicating that sweet clover seedlings increased chlorophyll pigments to synthesize photosynthetic products and thus improved saline–alkali tolerance [42]. When the saline–alkali solution concentration reached the maximum, in the salt-sensitive variety NXWF, the chlorophyll content decreased more than in 061898, these different amplitude changes reflecting the differential tolerance among sweet clover varieties.

Chlorophyll fluorescence can reflect plant photosynthetic physiology under saline–alkali stress [43], which leads to osmotic stress that can damage chloroplast membranes [44]. Lowered chlorophyll fluorescence parameter values indicated that PSII was damaged by saline–alkali stress. The higher the alkaline concentration and the lower the saline–alkali tolerance, the more severe damage to PSII. Salt stress reduces the Fv/Fm ratio, affecting the efficiency of photosystem II [45], which can indicate the degree of salinity stress suffered by plants [46]. Our observed decreases in chlorophyll fluorescence parameter values indicated that PSII was damaged by salt stress and that the plants were more affected by saline–alkali stress. In this study, the change in the Fv/Fm ratio before and after stress was greater than zero in 061898, 061930, and 55WF under $Na_2CO_3$ and $NaHCO_3$ stresses, as well as for all four sweet clover varieties under NaCl stress. However, the difference between the values before and after stress was smaller, indicating that sweet clover was able to effectively maintain PSII activity and potential maximum light energy conversion efficiency. Under saline–alkali stress, so, the plant light and reaction tend to stabilize, to prevent the net photosynthetic rate from decreasing too much [47]. This is a protective mechanism for sweet clover's adaptation to saline–alkali stress environments [23,48].

4.2.2. Effects of Saline–Alkali Stress on $H_2O_2$ and $O_2{}^-$ Accumulation and Cell Death

The accumulation of two important ROS elements ($H_2O_2$ and $O_2{}^-$) in leaves under saline–alkali stress conditions was observed by using DAB and NBT staining, reflecting the degree of plant leaf damage [23]. In this experiment, the levels of $H_2O_2$ and $O_2{}^-$ accumulation in the leaves of the four sweet clover varieties increased with the concentration of the saline–alkali solution, and at the maximum saline–alkali solution stress, 061898 accumu-

lated the lowest $H_2O_2$ and $O_2^-$. Higher accumulations of $H_2O_2$ and $O_2^-$ were observed in the severely damaged NXWF, which is in agreement with Senjuti Sen et al. [23] and An YM et al. [49]. In this case, the effect of alkaline salt was greater than that of neutral salt.

Salt and alkali stress lead to plant cell death, so plant cell activity detection is used as a direct and effective method to determine the extent of plant damage and to test salinity tolerance [24]. Consistent with the trend of ROS accumulation, the cell death area of saline–alkaline resistant variety 061898 was significantly smaller than that of NXWF at the highest concentration. These results are in accordance with Senjuti Sen et al.'s [23] findings on the salt stress of chickpea cell death, as well as Liu Nan et al.'s [24] research on how salt stress affects the cells in *Arabidopsis thaliana* leaves.

### 4.2.3. Effects of Saline–Alkali Stress on Osmoregulation MDA Content

Osmoregulation is one of the most basic features of plant salinity tolerance, and its enhancement is a crucial mechanism to increase plant salinity resistance [32]. The most common osmoregulators are free proline and soluble sugars [50,51]. Numerous studies have shown that the higher the content of soluble sugars accumulated by the plant in a certain period under adverse conditions, the higher the plant's resilience [52]. In this study, the leaf soluble sugar content of the four sweet clovers initially increased and then decreased with increasing saline–alkali solution concentrations. The saline–alkaline-resistant 061898 accumulated more soluble sugar. Same as soluble sugar, the proline content in the four sweet clover varieties exhibited an increase, and the increase in 061898 was greater than in NXWF. However, it has been observed that, beyond a certain concentration of saline–alkali, the intracellular concentration of proline starts to decrease. This could be due to increased saline–alkali causing lipid peroxidation, which could affect the permeability of the cell membrane, leading to an inability to accumulate proline.

MDA is an important end product of membrane lipid peroxidation, and its enrichment further reduces cell membrane stability [53], exacerbating membrane system damage and cellular senescence. High and low MDA levels directly reflect the degree of damage to the cellular membranes and resistance to adversity and stress conditions [54]. The study confirmed that salinity and alkalinity stress induced an increase in the MDA content in various plants [55–57]. A low saline–alkali concentration is less injurious to plants, and their defense system can efficiently fight against the cell damage caused by saline–alkali stress [57]. Under saline–alkali stress conditions, the MDA content in the more resistant sweet clover varieties increased significantly less than in the sensitive varieties, which is consistent with the findings of Li Ruiqiang et al. [48]. Among the three saline–alkali solutions, the degree of damage to membrane lipids inside plant cells was significantly greater with alkaline salt than with neutral salt.

### 4.2.4. Effect of Saline–Alkali Stress on Antioxidant Enzyme Activities (SOD, POD, CAT)

A membrane protective enzyme system capable of scavenging ROS exists in plants and mainly includes superoxide dismutase (SOD), peroxidase (POD), and catalase (CAT) [58]. These convert excessive intracellular ROS into harmless substances, such as $H_2O$, and maintain the balance of the reactive oxygen species metabolism in the plant body, to avoid damage by reactive oxygen species [59,60]. Changes in antioxidant enzyme activities show the degree of plant antioxidant damage, which is an important indicator of plant abiotic stresses [10]. In this study, SOD, POD, and CAT activities of sweet clover seedlings under the stress of the three saline–alkali solutions increased and then decreased with saline–alkali solution concentrations. When the concentration of the three saline–alkali solutions increased to a certain value, all four sweet clover enzyme activities decreased, indicating that the protective enzyme system balance was disrupted by increased saline stress, which exacerbated membrane lipid peroxidation and caused membrane damage [61]. Salt-tolerant varieties possessed higher antioxidant enzyme activity [11]. The resistant variety 061898 maintained relatively high enzyme activity than NXWF under saline–alkali stress, thus mitigating membrane damage caused by membrane lipid peroxidation. Meanwhile, SOD,

POD, and CAT activities in sweet clover under neutral salt stress were higher than under alkaline salt stress, and Yuan Lin et al. [5] similarly found that alkaline salt had a greater effect than neutral salt on the Pistachio protective enzyme system.

*4.3. Sweet Clover Gene Expression under Saline–Alkali Treatment*

Excessive ROS production can cause serious oxidative damage, while appropriate ROS production acts as a defense signal to improve stress tolerance [62]. ROS-activated mitogen-activated protein kinases (MAPKs), in combination with activated signal molecules, initiate several transcription factors like WRKYs, resulting in the transcription of multiple stress responsive genes [63–65]. Recent studies suggest that *AtWRKY33* may act as a direct or indirect regulator, coordinate or control downstream salt stress-related genes, and affect plant salt tolerance [66]. Our results indicate that in resistant sweet clover 061898, an upregulation in the expression level of WRKY33 occurred when treated with $Na_2CO_3$, whereas an increase in *WRKY33* expression was also detected in the salt-sensitive variety under mild saline–alkali stress. This may be associated with the accumulation of higher levels of ROS under alkali stress.

In Arabidopsis, an oxidative signal-inducible gene (OXI1) has been shown to function upstream of MAPK cascades [67]. An OXI1-MPK3/MPK6 cascade was activated by ROS [68] and OXI1-NDPK2-MPK3/MPK6 was suggested to enhance tolerance against freezing and salinity [69]. Additionally, the mitogen-activated protein kinase (MAPK2) kinase 2 (MKK2) and the downstream MPK4 and MPK6 were shown to be key regulators of ROS stress signaling [70,71]. In Arabidopsis, over-expression of wild-type or constitutively active MKK2 resulted in elevated MAPK kinase activity of MKK2 and enhanced salt tolerance [71]. We found that in 061898, *MKK2* expression level downregulated, especially in $Na_2CO_3$ treatment, and *OXI1* and *MPK3* were significantly upregulated in response to NaCl and $Na_2CO_3$ treatments. For NXWF, *OXI1* and *MPK3* expression levels notably increased after $NaHCO_3$ stress. There was a corresponding increase in leaf ROS accumulation. The activation of ROS triggered *OXI1* and *MPK3*, enabling sweet clover to adapt to high saline–alkali environments. Yet, we noted variations in the expression profiles between resistant and susceptible genotypes, as well as differences in response to various types of saline–alkali conditions.

Hormones are major players in the establishment and interconnection of plant signaling networks and have traditionally been divided into two groups: growth-related hormones (such as auxin, gibberellic acid (GA), cytokinins, brassino steroids, and strigolactones) and stress-related hormones (such as abscisic acid (ABA), salicylic acid (SA), ethylene (ET), and JA) [72–74]. We quantified the expression levels of genes associated with signaling pathways for ABA (*PYL*, *PP2C*), JA (*COI1*, *JAZ*, *MYC2*), SA (*GH3*, *TGA*), and BR (*CYCD3*). The gene exhibited distinct expression patterns, suggesting that both resistant and susceptible sweet clover genotypes possess a certain level of tolerance to NaCl, with the JA signaling pathway likely serving as a primary modulator regulating sweet clover saline–alkali tolerance. In susceptible genotypes, in addition to the JA pathway, the SA pathway may also play a role in the response to mild alkaline conditions.

**5. Conclusions**

In conclusion, to alleviate the damage caused by saline–alkali stresses, sweet clover seedlings enhanced their stress resistance by elevating the content of osmoregulatory substances, enhancing the activity of antioxidant enzymes, and modulating gene expression levels to adapt to the stressful environment. Sweet clover's tolerance to three types of salts is ranked as follows: NaCl > $NaHCO_3$ > $Na_2CO_3$. Salt-alkali resistant sweet clovers exhibit the capacity to sustain normal growth under high saline–alkali stress, positioning them as promising candidates for the reclamation of saline–alkaline soil. Furthermore, this study uncovers differential gene regulatory patterns and signaling pathways under neutral and alkaline salt stresses. A deeper exploration of these molecular responses to individual salt stresses could facilitate a more comprehensive understanding of the complex control

mechanisms operative under combined salt stress conditions. Our study offers valuable guidance for the exploitation and utilization of sweet clover, as well as for the breeding of new cultivars.

**Author Contributions:** Conceptualization, Z.W., J.Y., Z.S. and C.G.; methodology, Z.W. and J.Y.; software, Z.W., J.Y. and X.Z.; validation, Z.W., J.Y., X.X., Y.Y., J.W., D.Z. and L.M.; resources, J.W. and D.Z.; writing—original draft preparation, Z.W. and J.Y.; funding acquisition, Z.S. and C.G. All authors have read and agreed to the published version of the manuscript.

**Funding:** This research was funded by "Agricultural Technology Innovation Project of Heilongjiang Province, CX23GG06", "Scientific Research Foundation of Heilongjiang Province, CZKYF2022-1-B012", "Research Foundation of Heilongjiang Academy of Agricultural Sciences (2020YYYF009)", and "National Natural Science Foundation of China (NSFC) (U21A20182)".

**Data Availability Statement:** Data are contained within the article.

**Conflicts of Interest:** The authors declare no conflicts of interest.

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
