# Peer review of "Physiological and Biochemical Responses of Melilotus albus to Saline and Alkaline Stresses"

_horticulturae, doi:10.3390/horticulturae10030297_

Round 1

Reviewer 1 Report

Comments and Suggestions for Authors

The reviewed study concerns the response to salinity stress in various varieties of Melilotus albus. Although the paper is well-written, there is a lack of emphasis on the new information derived from the conducted research. The conducted analyses are quite rudimentary, and the observed relationships are consistent with previously published research. Consequently, I suggest a substantial reduction in the overall length of the paper.

Detailed comment:

- All figures in their current dimensions are illegible;

- The discussion contains numerous repetitions from the results section;

- The chosen keywords are quite generic;

- When indicating the percentage increase/decrease of a given value, it is preferable to provide whole numbers, as specifying, for example, 50.24%, does not significantly enhance precision beyond 50%

- From the conclusions, I failed to glean any novel insights from the entire study; try to improve them;

The remaining comments are included in the PDF file. The highlighted phrases should be revised, or I request the authors to address the specific comment.

Reviewer 2 Report

Comments and Suggestions for Authors

I read with interest the manuscript entitled “Physiological and Biochemical Responses of Melilotus albus to Different Saline and Alkaline Stresses”. This study aimed to select four varieties of sweet clover with different sensitivities (061898, 061930, No. 55 white flower, and Ningxia white flower), and analyzed the effects of different concentrations of three sodium salts (sodium carbonate, sodium bicarbonate, and sodium chloride) on their physiology and biochemistry. The subject of the article is important and has great relevance for the scientific environment of the study area. Therefore, the manuscript needs some adjustments so that it can then be forwarded to the publication process. The manuscript has the potential for publication in this journal Horticulturae and needs the following adjustments:

TITLE

- Delete the word “different”.

ABSTRACT

- The objective described here must be the same as that described in the Introduction. To check.

- Replace repeated keywords in the title. All.

INTRODUCTION

- Why is the name of the species mentioned in the first paragraph?

- Reduce paragraphs. There is a lot of information that makes it difficult for the reader to understand.

- Does the Introduction only have two paragraphs?

- Enlarge this section. Add more information and separate paragraphs. Review this entire section.

MATERIAL AND METHODS

- The methodology needs to be more detailed.

- Enter more information about the study location.

- The experimental design needs to be described in a better way.

- Was the experiment in the field or greenhouse?

- This section needs to be improved a lot.

RESULTS AND DISCUSSION

- All graphics need to be improved. It's difficult to visualize.

- Figure captions must be self-explanatory. It is necessary to explain what the figure means. For example, Figure 8 and others.

CONCLUSIONS

- Reduce. To check.

Reviewer 3 Report

Comments and Suggestions for Authors

The topic of the article is very interesting, but it has shortcomings.

1. Melilotus albus - it is not a desirable plant because it quickly becomes woody and loses its usefulness.

2. Seed hardness - why sandpaper? There is a lot of error because the seeds for sowing may be damaged in various ways. where's the scarifier?

3. A pot measuring 11x11 cm, what is its depth?

 4. How many repetitions?5. Physiological measurements on how many plants?

6. The summary lacks typical agricultural recommendations

 7. The drawings are poorly visible and need to be improved

 8. List of literature, too many scientific articles published over 20 years earlier

 To sum up, in my opinion, the article contains a serious methodological error regarding damage to the seed coat - the decision should be rejected

Round 2

Reviewer 2 Report

Comments and Suggestions for Authors

Dear,

The authors made the suggestions proposed in the previous version.

My only suggestion is to increase the font size of all figures. This way it is not easy to see.

After this correction, the article has the potential to be published in this journal.

Reviewer 3 Report

Comments and Suggestions for Authors

I accept all changes and replies.
